# Engineering protein-protein devices for multilayered regulation of mRNA translation using orthogonal proteases in mammalian cells

Federica Cella [1,2], Liliana Wroblewska[3], Ron Weiss[4] & Velia Siciliano [1]

The development of RNA-encoded regulatory circuits relying on RNA-binding proteins (RBPs) has enhanced the applicability and prospects of post-transcriptional synthetic network for reprogramming cellular functions. However, the construction of RNA-encoded multilayer networks is still limited by the availability of composable and orthogonal regulatory devices. Here, we report on control of mRNA translation with newly engineered RBPs regulated by viral proteases in mammalian cells. By combining post-transcriptional and post-translational control, we expand the operational landscape of RNA-encoded genetic circuits with a set of regulatory devices including: i) RBP-protease, ii) protease-RBP, iii) protease–protease, iv) protein sensor protease-RBP, and v) miRNA-protease/RBP interactions. The rational design of protease-regulated proteins provides a diverse toolbox for synthetic circuit regulation that enhances multi-input information processing-actuation of cellular responses. Our approach enables design of artificial circuits that can reprogram cellular function with potential benefits as research tools and for future in vivo therapeutics and biotechnological applications.

[1] Istituto Italiano di Tecnologia-IIT, Largo Barsanti e Matteucci, 80125 Naples, Italy. [2] University of Genoa, 16132 Genoa, Italy. [3] Biomedicine Design, Pfizer Inc, Cambridge, MA 02139, USA. [4] Synthetic Biology Center, Department of Biological Engineering, Massachusetts Institute of Technology, 500 Technology Square, 02139 Cambridge, MA, USA. Correspondence and requests for materials should be addressed to V.S. (email: velia.siciliano@iit.it)

Mammalian synthetic biology is developing research tools and programmable therapeutics that are based on sophisticated genetic circuit design[1,2]. To date, most devices used in synthetic biology are encoded by DNA, with associated risks of insertional mutagenesis and immunogenicity[3]. RNA-based circuits overcome these limitations while still providing control over strength, timing, and location of gene expression[4,5]. In addition, RNA does not need to cross the nuclear barrier and can be delivered into non-dividing, as well as dividing cells[3], with advantages for broad biomedical, industrial, and pharmaceutical applications including cancer immunotherapy[6], genome engineering, and genetic reprogramming[7]. The development of foundational tools that control RNA activity is thus paramount to engineering RNA-encoded synthetic networks to program cellular functions.

Initial efforts on the design and characterization of new genetic modules to implement RNA-based networks focused on building parts such as aptamers, ribozymes, and riboswitches[8–11]. These can modulate the translation of associated output, but cannot currently be interconnected to implement complex circuits. The ability to compose parts into circuits has recently become possible with the use of RNA-binding proteins (RBPs) such as L7Ae and MS2-cNOT7. These RBPs have been successfully interconnected to create post-transcriptional circuits in which they function both as inputs and outputs of the regulatory devices[12]. Here, we propose to further expand the available toolset of post-transcriptional regulators using novel protein–protein interaction-based devices through re-engineering of RBPs and proteases.

Protein systems engineering using well-characterized standardized protein modules offers several useful features such as enabling information processing and actuation in target cells and in vitro optimization before investigating their behavior in vivo[13]. Proteolysis devices present a possible platform for multi-element circuit engineering, but today are largely used in synthetic biology to modulate single device protein half-life, or to trigger selected cellular responses[14–17]. Proteases are broadly expressed in all organisms and function by cleaving short, specific amino acid sequences. Proteases are involved in all areas of metabolism and constitute key effectors of several cellular pathways, modulating protein–protein interaction, protein localization, and ultimately regulating cellular fate[18,19]. They are classified on the basis of cleavage site[20,21] (i.e., endopeptidases cleave the target protein internally and exopeptidases remove single amino acids from N or C-terminal end of the protein), evolution, substrate specificity, and reaction catalyzed[22,23]. Proteases can be adapted to operate in several environmental conditions and optimized to function in different organisms. Viral proteases such as tobacco etch virus protease (TEVp) are widely used in biotechnology for endoproteolytic removal of affinity tags from recombinant proteins[24], and in synthetic biology to help convert extracellular or intracellular ligand sensing into transcriptional output[15–17,25]. They are also used to control protein levels via an N-terminal degron fused to the target protein with a protease-responsive cleavage site, as demonstrated in bacteria[14]. To date there are no off-targets substrates for TEVp reported in the human proteome[26] and in other organisms where they are in fact well tolerated[27–29]. The use of TEVp has enabled innovative studies of biological processes. For example, engineering of a TEVp responsive Rad21 in a cell-type-specific and temporally controlled manner unveiled cohesin function in post-mitotic cells like Drosophila neurons[27]. We thus envisioned that proteases represent an attractive tool for engineering more sophisticated post-transcriptional and translational regulatory networks, and that protease-RBP and protease–protease regulatory devices could lay the foundation for circuits that are solely RNA-encoded and are based on protein–protein interactions.

Toward this goal we inserted protease cleavage sites into RBP and proteases sequences, linking their activity to a set of upstream proteases. The devices were genetically encoded on plasmid DNA to allow rapid testing in human embryonic kidney (HEK293FT) cells via transient transfections.

As discussed below, we first engineered L7Ae to harbor a TEVp cleavage site (TCS), and implemented a set of circuits (an intrabody-based protein sensor, a cascade, and a switch) that operate solely with protein:RNA (L7Ae:2Kt) and protein:protein (TEVp:L7Ae) interactions. Next, we inserted into MS2-cNOT7 specific cleavage sites for any of four proteases: TEVp, tobacco vein mottling virus protease[30] (TVMVp), turnip mosaic virus protease[31] (TUMVp), and sunflower mild mosaic virus protease (SuMMVp), and implemented post-transcriptional regulatory cascades. Finally, we functionally connected proteases to create multi-layered cascades.

Our work expands the toolbox of parts available for post-translation control, and provides an important proof of concept study that enables new means to control post-transcriptional regulatory circuit behavior with endogenous or exogenous protein inputs that respond to cellular and environmental changes.

## Results

**Engineering TEV protease-responsive L7Ae.** The archaeal RNA-binding protein L7Ae binds the RNA C/D box that forms a classical kink-turn (K-turn) motif[32,33] in the 5'UTR of target mRNAs, inhibiting their translation[34,35]. As a first step to link L7Ae activity to TEVp, we identified possible insertion points for the TEV cleavage site (TCS) based on the crystal structure of L7Ae bound to its native RNA recognition sequence. We had two main objectives in choosing the sites: (i) TCS insertions should minimally affect L7Ae structure or RNA-binding, and (ii) TEVp cleavage should render L7Ae non-functional. We thus chose three candidate insertion sites for TCS in loop regions away from the K-turn binding domain and closer to the center of the L7Ae sequence (Fig. 1a, Supplementary Figure 1). Accordingly, we built and tested three new translation repressors (L7Ae-CS1, L7Ae-CS2, L7Ae-CS3) along with a reporter gene regulated by two repeats of the K-turn motif in its 5'UTR (2Kt-dEGFP). As shown in Fig. 1b, in the absence of TEVp (State 1) and similar to the behavior with wild-type L7Ae, the three L7Ae-CS repressors downregulate translation of the reporter gene. In the presence of TEVp (State 2 in Fig. 1b), we observed 13.4 fold derepression of CS2 and 77 fold derepression of CS3, but no appreciable response to CS1 (Fig. 1c, Supplementary Figures 2, 3, 18). The circuit harboring L7Ae-CS3 showed efficient and sustained repression/derepression over a long-time frame (96 h) (Supplementary Figure 4) and in another mammalian cell line (HeLa cells, with slightly lower repression efficiency; Supplementary Figure 5). We also tested additional flexible linker sequences flanking the CS or multiple insertions within L7Ae but did not observe any significant improvement to the initial set of repressors. As discussed below, we used this new protease-dependent translation system to create several circuits, including an HCV protein sensor, a cascade, and a switch.

**L7AeCS3-based regulatory networks.** The HCV sensing device detects NS3, a serine protease, which is considered an attractive target for HCV therapy as it is responsible for processing the polyprotein precursor encoded by the virus and is essential for viral replication[36]. We used two single-chain fragment intrabodies scFv35 and scFv162 specific for two distinct epitopes of NS3 previously reported to interfere with its viral activity[37]. We have recently shown that these intrabodies can be efficiently wired into a transcriptionally based protein-sensing-actuation

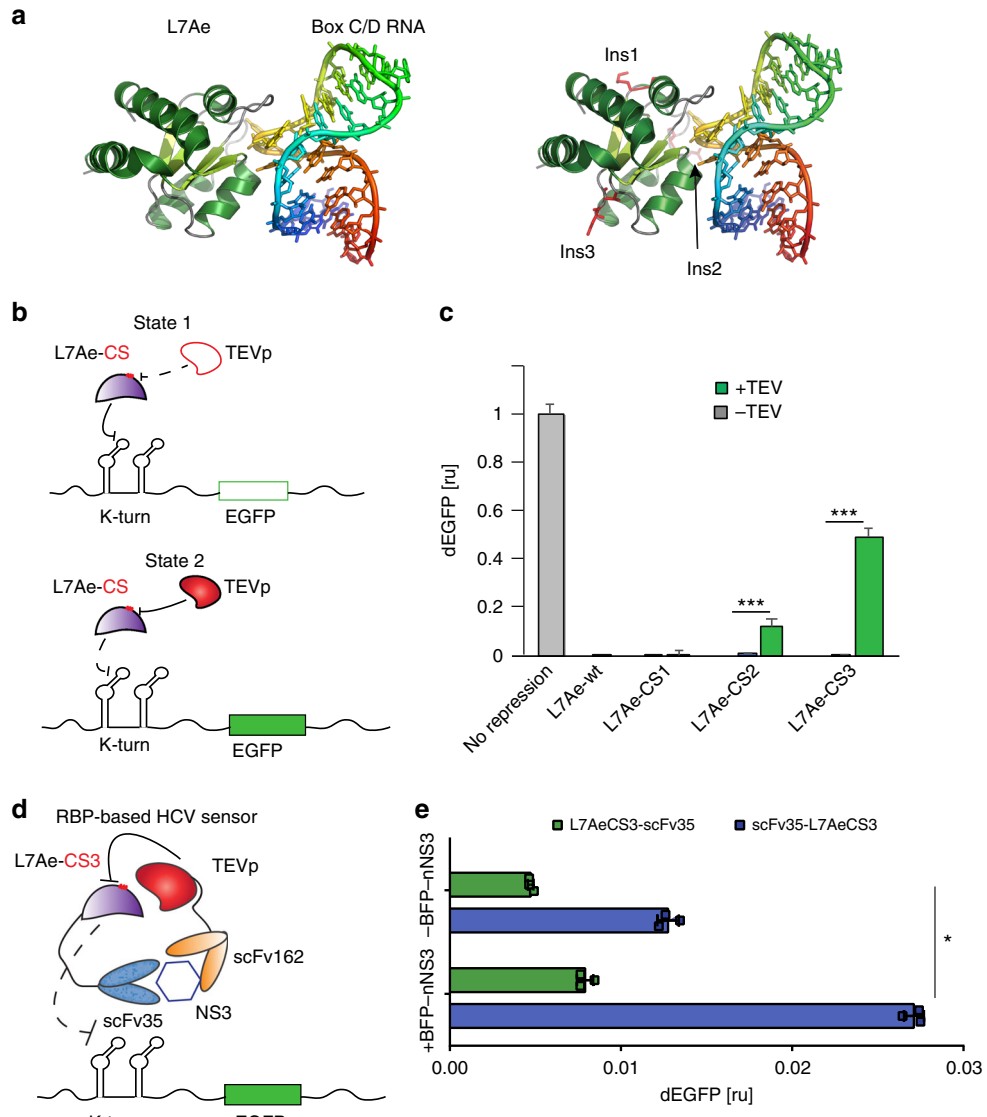

**Fig. 1** Engineering a TEVp responsive L7Ae. **a** Structure of L7Ae binding to box C/D RNA that forms a K-turn motif. TCS insertion sites in L7Ae were placed after amino acid residue N70 (L7Ae-CS1), P56 (L7Ae-CS2), or K77 (L7Ae-CS3), shown in red for each site. Protein structure was determined in[33] (PDB id: 1RLG). L7Ae-CS structure visualization was performed with pymol[65]. **b** Schematics of L7Ae with inserted TCS translational regulation. In the absence of TEVp, L7Ae-CS binds and repress the K-turn motifs in the 5′UTR of mRNA target (EGFP OFF-State 1). When TEVp is expressed, it cleaves the TCS, disrupting L7Ae structure and inhibiting its function (EGFP ON-State 2). **c** Flow cytometry analysis of three engineered L7Ae-CS tested in HEK293 cells in the absence or presence of TEVp. Data represent geometric mean of dEGFP normalized by transfection marker mKate expressed from the same constitutive promoter to account for different expression across the cell lines. ru, relative units. Significant changes determined by unpaired $t$-test are indicated with asterisks ***$p$-value < 0.001 $n$ = 3 replicates. **d** Schematics of L7Ae-CS3-based HCV sensor. TEVp-scFv162[16] and L7AeCS3-scFv35 fusion proteins are modules of a sensor for NS3 protein, which is associated with HCV virus. L7Ae-CS3 represses the mRNA target in the absence of NS3. In the presence NS3, binding of the intrabodies brings TEVp in proximity to L7Ae-CS3, resulting in cleavage of the RBP and derepression of the target gene. **e** Test of the variants of NS3 sensor in the presence and absence of BFP-nNS3[16]. Data represent geometric mean and standard deviation of means of dEGFP normalized by transfection marker mKate, for $n$ = 3 replicates. Significant changes determined by unpaired $t$-test are indicated with asterisks *$p$-value < 0.05. Data collected 48 h post transfection

device to detect NS3 expression in mammalian cells[16]. Here, we designed a new device in which the sensor is based on translational regulation using L7Ae-CS3 and TEVp fused to scFv35 and scFv162, respectively (Fig. 1d). The presence of the target protein and subsequent binding of the two intrabodies should result in elevated TEVp cleavage of the RBP and derepression of 2Kt-dEGFP (Fig. 1d). Similar to what we have shown previously[16], to achieve specific output expression in the presence of NS3 and minimize non-specific cleavage by TEVp, we regulated protease expression with a promoter inducible by doxycycline[16]. We have

previously demonstrated that TEVp fused to scFv162 retains protease activity[16]. Here, we tested scFv35 intrabody fused to N-terminus or C-terminus of wild-type L7Ae (wL7Ae) or L7AeCS3. We observed efficient dEGFP repression by all chimeric proteins (Supplementary Figure 6a, b), and both scFv35-L7Ae-CS3 and L7Ae-CS3-scFv35 respond to TEVp, rescuing reporter translation (Supplementary Figure 6b, c). Next, we tested the HCV sensor in HEK293 cells by expressing scFv35-L7Ae-CS3 or L7Ae-CS3-scFv35 along with TEVp-scFv162, in the presence or absence of a blue fluorescence-tagged NS3 (BFP-nNS3)[16], useful for live cell

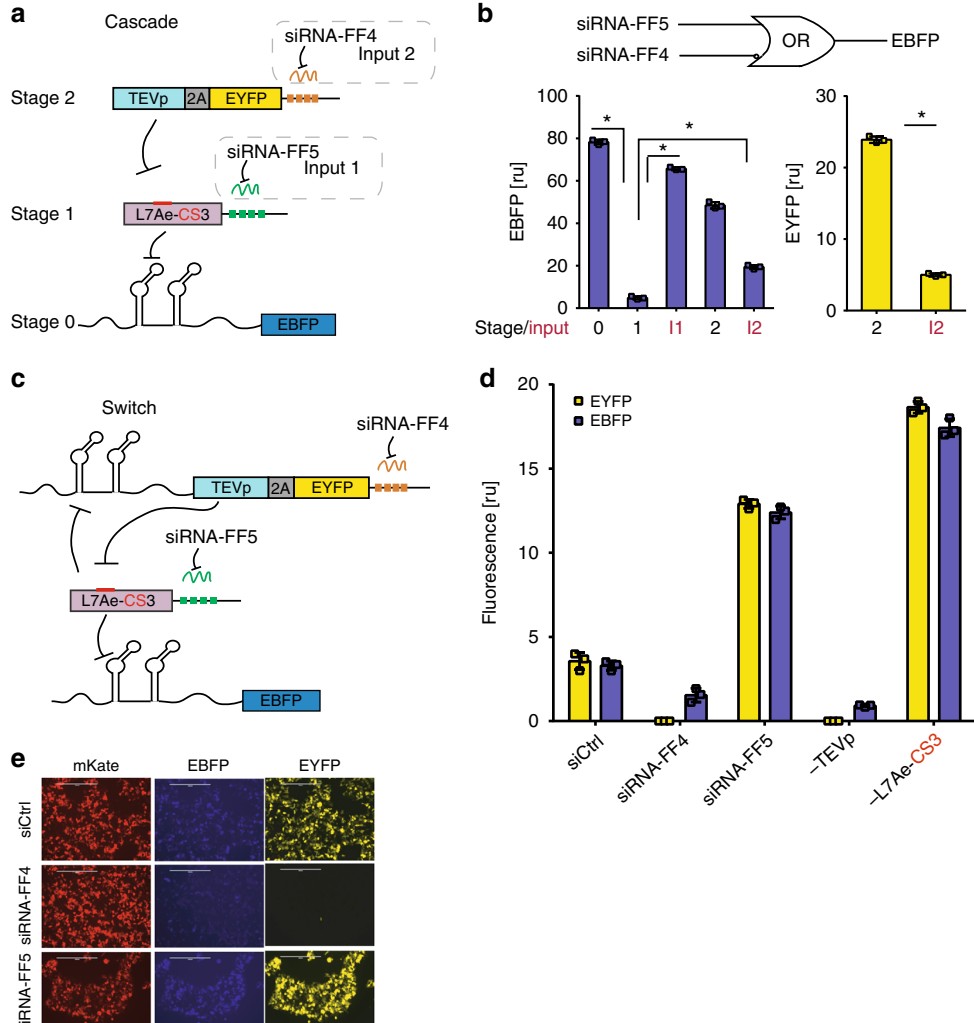

**Fig. 2** L7Ae-CS3-based cascade and two-state switch. **a** The cascade includes both protein–protein regulation and post-transcriptional regulation by siRNA. The L7Ae-CS3 transcript incorporates four siRNA-FF5 target sites in the 3'UTR. TEVp, which is co-expressed with EYFP, is downregulated by siRNA-FF4 via target sites in its 3'UTR. EBFP is expressed in absence of L7AeCS3 (Stage 0) and repressed in the presence of the RBP (Stage 1). TEVp cleaves L7Ae-CS3, rescuing EBFP expression (Stage 2). siRNA-FF5 (Input 1) downregulates L7Ae-CS3 in the absence of TEVp. siRNA-FF4 (Input 2) knock down of TEVp results in high L7Ae-CS3 levels and thus in EBFP repression. **b** Simplified logic circuit and corresponding flow cytometry data. The cascade implements the two-input logic "siRNA-FF5 OR (NOT siRNA-FF4)" (top). Bar charts of the EBFP and EYFP circuit outputs as additional stages of the cascade and the inputs are added to the experiment, starting from just the reporter (Stage 0), and then adding L7Ae-CS3 (Stage 1), input siRNA-FF5 (Input 1) or TEVp-2A-EYFP (Stage 2), and siRNA-FF4 (Input 2). Data represent geometric mean and standard deviation of means of EBFP and EYFP normalized by transfection marker mKate for $n = 3$ replicates. ru relative units. Significant changes determined by unpaired $t$-test are indicated with asterisks. *$p$-value < 0.05. **c** Schematics of the L7Ae-CS3-based switch. TEVp-2A-EYFP includes two K-turn motifs in the 5'UTR and four target sites for siRNA-FF4 in the 3'UTR. L7Ae-CS3 mRNA harbors four siRNA-FF5 target sites in the 3'UTR. To monitor L7Ae-CS3 activity, we included EBFP reporter with two K-turn motifs in the 5'UTR. siRNA-FF4 and siRNA-FF5 are used to set and reset the state of the switch. **d** Flow cytometry data representing network behavior. Data represent geometric mean and standard deviation of means of EBFP and EYFP normalized by transfection marker mKate for $n = 3$ replicates. ru relative units. **e** Representative fluorescent micrographs of the switch showing the state of the switch regulated by siRNA-FF4 and siRNA-FF5. Scale bars indicate 200 μm. Data collected 48 h post transfection

monitoring of target protein expression. dEGFP expression increased in the presence of BFP-nNS3, indicating specific protein-sensing cleavage of L7Ae/TCS variants by TEVp. Of note, we observed stronger derepression with scFv35-L7Ae-CS3, perhaps due to facilitated interaction with the TEVp cleavage site, and in the absence of doxycycline, indicating that leakiness of the promoter was sufficient to induce TEVp-mediated cleavage, in accordance with previous observations[16] (Fig. 1e).

To explore multi-input information processing and actuation of cellular responses, we next designed a two-input three-stage repression cascade with an overall logic function that is true in

the presence of siRNA-FF5 or absence of siRNA-FF4. This circuit operates using both post-transcriptional and post-translational regulation (Fig. 2a, Supplementary Figure 7). The first input to the cascade is the synthetic short interfering RNA FF4 (siRNA-FF4) that downregulates expression of TEVp through four repeats of the FF4 target site encoded in the 3'UTR. To monitor TEVp down-modulation, we designed TEVp-2A-EYFP that couples fluorescence intensity to protease expression through a self-cleaving 2A peptide. At the next stage of the cascade, TEVp cleaves L7AeCS3, which otherwise binds K-turn motifs in the 5'UTR of RNA encoding a blue fluorescent protein (2Kt-EBFP)

(Fig. 2a). For the second input, we placed four repeats of the target site for the synthetic short interfering RNA FF5 (siRNA-FF5) in the 3'UTR of L7AeCS3. We tested the behavior of the circuit in HEK293 cells and observed that EBFP is repressed by L7AeCS3 (stage 1), whereas knocking down L7AeCS3 activity either by siRNA-FF5 (input 1) or by TEVp (stage 2) restored reporter output. siRNA-FF4 (input 2) induced TEVp down-regulation, and resulted in rescue of L7AeCS3 activity and EBFP repression (Fig. 2b).

Then, we engineered a switch based on protein–protein/RBP-RNA cross regulation, in which TEVp and L7AeCS3 repress each other (Fig. 2c) with a general cross repression topology resembling previously described mammalian and bacterial switches[12,38,39]. The switch is similar to the cascade, with the addition of two K-turn motifs in the 5'UTR of TEVp-2A-EYFP (Fig. 2c). Administration of either siRNA-FF4 and siRNA-FF5 is used to set or reset the state of the switch. In the absence of siRNAs, both repressors and reporter genes remained at intermediate levels 48 h post-transfection (Fig. 2d, Supplementary Figure 8). siRNA-FF4 sets the state to high L7AeCS3 resulting in low EYFP and EBFP, whereas siRNA-FF5 resulted in the opposite state (Fig. 2d, e, Supplementary Figure 8).

**Engineering protease-responsive MS2-cNOT7 repressor devices.** MS2-cNOT7 fusion RBP targets the 3'UTR of mRNAs containing MS2-binding motif and catalyzes mRNA de-adenylation[12]. We first inserted a TCS at the junction of MS2-cNOT7 (MS2-TCS-cNOT7) to enable regulation by TEVp (Fig. 3a). In the absence of TEVp, MS2-TCS-CNOT7 showed similar repression to its wild-type counterpart, whereas co-expression of the protease fully restored EGFP expression, indicating efficient and sustained disruption of RBP activity (Fig. 3b, Supplementary Figures 9, 10). We also confirmed that wild-type MS2-cNOT7 activity is not affected by TEVp (Supplementary Figure 10). Next, we examined whether we could re-engineer the synthetic RBP to respond to different proteases. We therefore built a library of MS2-CS-cNOT7 variants responsive to SuMMVp[14], TUMVp, and TVMVp[30] proteases. For these, we tested two alternative cleavage sites (Serine-S or Glycine-G in P1[40] position of the cleavage sequence) with different affinities for the cognate protease (Fig. 3c). We did not include TEVp since previous experiments performed in our lab showed similar response when using either of the two cleavage sites. All MS2-CS-CNOT7 variants that we constructed repress EGFP translation (MS2-SuCS-cNOT7 exhibiting the strongest downregulation), and the cognate proteases disrupt this repression (highest efficiency obtained with TEVp and TUMVp), resulting in increased EGFP expression (Fig. 3b, d, Supplementary Figure 19). The three-vector system with proteases and associated RBPs encoded on separate plasmids was then used to measure crosstalk between non-cognate pairs. All combinations of MS2-CS-cNOT7 and proteases were co-transfected in HEK293 cells, demonstrating that the proteases are mostly orthogonal and therefore suitable to multi-layered circuit constructions (Fig. 3e, Supplementary Figures 11–14).

We hypothesized that we could extend our approach by designing proteases that include a cleavage site for an orthogonal protease, enabling a novel protein–protein regulation system. Towards this goal, we first designed TVMVp with three alternative insertion sites for the TUMVp cleavage site: (i) TVMVp-TUCS1 between amino acid residues D26-G27, (ii) TVMVp-TUCS2 between amino acid residues Q119-K120, and (iii) TVMVp-TUCS3 between amino acid residues T173-N174 (Supplementary Figure 15a). We tested the three variants along with MS2-TVCS-cNOT7 (MS2-cNOT7 with cleavage site

responsive to TVMVp), which in turn represses EGFP translation (Supplementary Note 1, Fig. 4a). Expression of TUMVp should result in TVMV-CS inhibition and restore EGFP down-modulation by MS2-TVCS-cNOT7 (Fig. 4a, Supplementary Figure 15a). We observed that one variant, TVMVp_TUCS2, interferes with MS2-TVCS-cNOT7 activity, while expression of TUMVp re-establishes EGFP translation repression (Fig. 4b, Supplementary Figures 16a, b, 20). Since TEVp and TVMVp share a high degree of structural similarity, we inserted TVMV-cleavage site (TVMV-CS) in the same amino acid regions, creating new TEVp-TVCS proteases (Fig. 4c, Supplementary Note 1, Supplementary Figure 15b). The TEVp-TVCS1 variant efficiently impairs MS2-TCS-cNOT7 repression of EGFP, and addition of TVMVp inhibits TEVp-TVCS1 function (Fig. 4d, Supplementary Figures 17, 20). As discussed in Supplementary Note 1, our results suggest an efficient insertion site within homologous proteases (amino acid region NFQX-KS, where X refers to variable amino acidic residue). Based on this information we designed a structurally modified TUMVp with TEVp cleavage site insertion (TCS) (Fig. 4e). Since the TUMVp crystal structure is yet unresolved, we instead inferred its structure based on homology to TEVp using SWISS-MODEL[41], an automated protein homology modeling server (Supplementary Figure 15c). We observed MS2-TUCS-cNOT7 inhibition by TUMV-TCS, with TEVp reverting the effect, albeit with reduced efficiency as compared to the other proteases (Fig. 4f, Supplementary Figure 20).

## Discussion

Synthetic devices that can be encoded by RNA are a promising framework to reprograming cellular function. These have potential applications both as research tools and for future in vivo therapeutics due to their transient expression and lack of chromosomal integration hazards, overcoming safety issues associated with DNA based circuit delivery. Indeed, a key aspect to consider when using RNA-encoded devices for biomedical applications is mRNA stability; chemical and structural modifications of in vitro transcribed mRNA have enabled production of significant levels of target proteins for durations longer than a week[7,42,43]. Recently, self-replicating RNA viruses such as Sindbis virus have garnered significant interest for a wide range of medical applications including vaccine delivery, gene therapy, and cellular reprogramming[44,45]. A main advantage of RNA replicons is that since they self-replicate, they can generate high levels of gene products even when starting from a relatively low dose, while the risk of undesired chromosomal integration is minimal because they do not reverse transcribe. Sindbis replicons with reduced cytopathicity and longer-term expression have been recently developed, making them an attractive platform for engineering RNA-encoded synthetic devices[46–48].

In this proof-of-concept study we lay the foundation for protein-driven synthetic circuits that enable construction of tunable RNA-encoded networks. Our approach relies on *Potyvirus* proteases. This class of proteins is attractive for many applications since to our knowledge they have no reported substrates in the human proteome[26] and in a broad range of other organisms that tolerate their expression[27–29]. Furthermore, the remarkable specificity of proteases to the 7-mer cleavage sequence (i.e., ENLYFQ/S recognized by TEVp[49]) minimizes the likelihood of unwanted knockdown of RBP function by endogenous proteases, thus creating an effective orthogonal system.

We demonstrated a set of regulatory networks including cascades, switch and protein sensor-actuator, showing that they operate in an orthogonal fashion combining post-transcriptional and post-translational regulation. We first engineered RBPs L7Ae

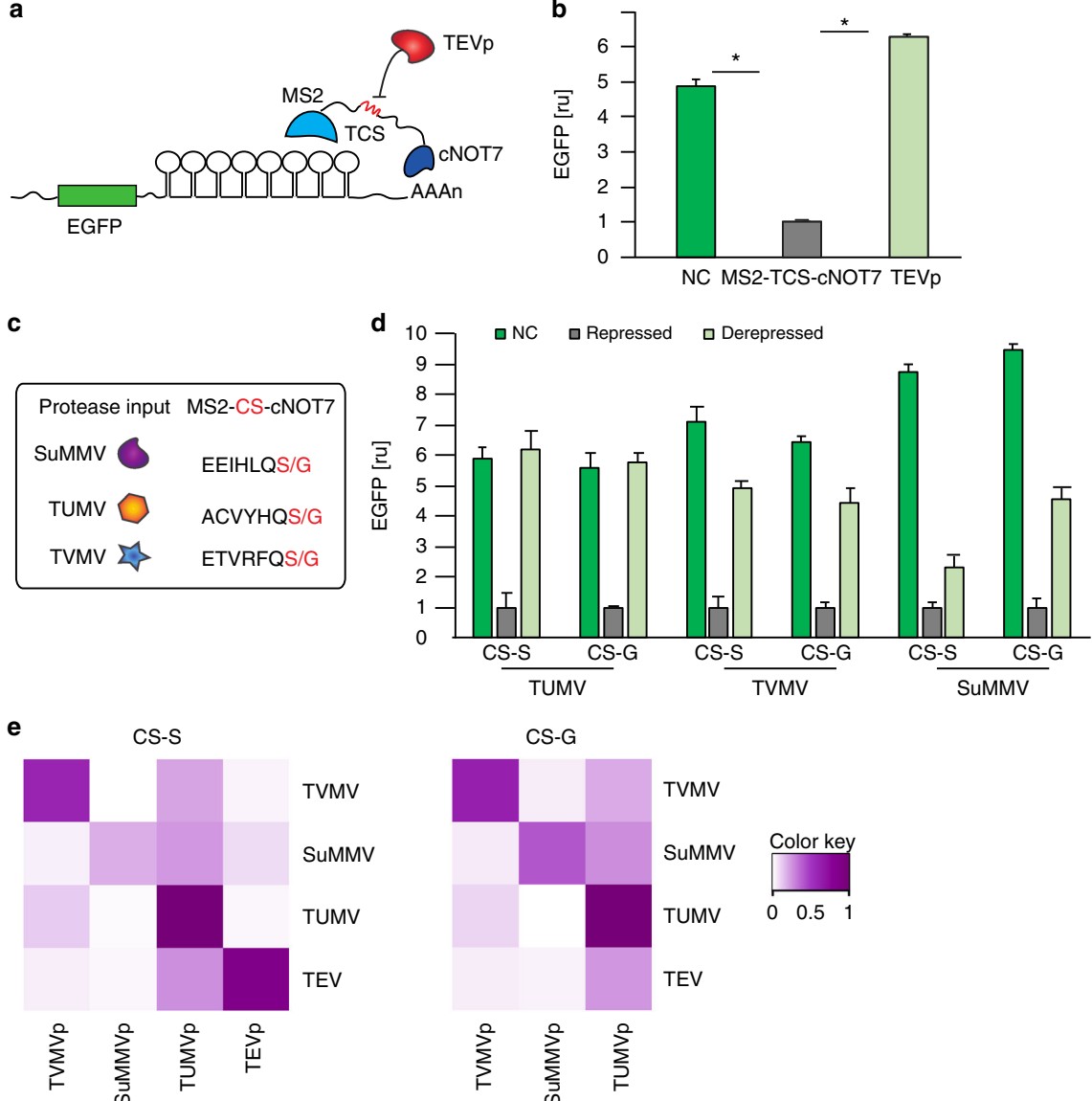

**Fig. 3** Engineering protease-dependent MS2-cNOT7 proteins. **a** Schematics of the cascade. MS2 is fused to cNOT7 via the TEVp cleavage site (MS2-TCS-cNOT7). MS2 binds its cognate sequences in the 3'UTR of target EGFP mRNA, resulting in RNA de-adenylation by cNOT7. When TEVp is co-expressed, MS2-TCS-cNOT7 is cleaved and the two domains are separated, rescuing EGFP translation. **b** Flow cytometry data of the protease cascade. MS2-TCS-cNOT7 shows repression similar to its wild-type counterpart MS2-cNOT7 (Supplementary Figure 10). EGFP translation recovers in the presence of TEVp. NC: negative control (no repression). Data represent geometric mean and standard deviation of EGFP normalized by transfection marker mKate for $n = 3$ replicates. ru relative units. Significant changes determined by unpaired $t$-test are indicated with asterisks *$p$-value $< 0.05$, $n = 3$ replicates. **c** Proteases and cognate cleavage site sequences used to create additional regulatory MS2-CS-cNOT7 proteins. **d** Analysis of MS2-CS-cNOT7 variants responsive to SuMMVp, TVMVp, TUMVp in the absence (Repressed) or presence (Derepressed) of the protease. Cleavage sites include either a Serine or a Glycine in P'1 with reported high affinity toward the associated proteases. The proteases are ordered from most efficient to least in terms of derepression of EGFP (TUMV > TVMV > SuMMV). Data represent geometric mean and standard deviation of EGFP normalized by transfection marker mKate for $n = 3$ replicates. NC: negative control (no repression). ru relative units. **e** Proteases orthogonality. Regulatory effect is shown between all combination of proteases and MS2-CS-cNOT7, for cleavage sites with Serine in P1 (left) and with Glycine in P1 (right). Data represent geometric mean of EGFP normalized by transfection marker mKate for $n = 3$ replicates. Data collected 48 h post transfection

and MS2-cNOT7 to include cleavage sites for cognate proteases, which exhibit robust and long-term protease-dependent regulation of target mRNA. With MS2-cNOT7 fusion protein, we created a protease-dependent system via cleavage sites inserted at the junction of the two modules. The design of L7Ae was based on analysis of the crystal structure of the RBP bound to native recognition RNA sequence, with aims to minimize alteration of wild-type L7Ae structure upon insertion, to minimize

interference with L7Ae's ability to bind RNA, and maximize cleavage in presence of TEVp that render L7Ae non-functional. The L7Ae proteins with integrated cleavage sites were used to compose three different devices, including NS3 protein sensor. We used an inducible promoter to control TEVp-scFv162 expression to minimize the likelihood of random cleavage[16]. Tuning gene expression post-transcriptionally could be achieved in a variety of ways, for example by adding a degradation domain

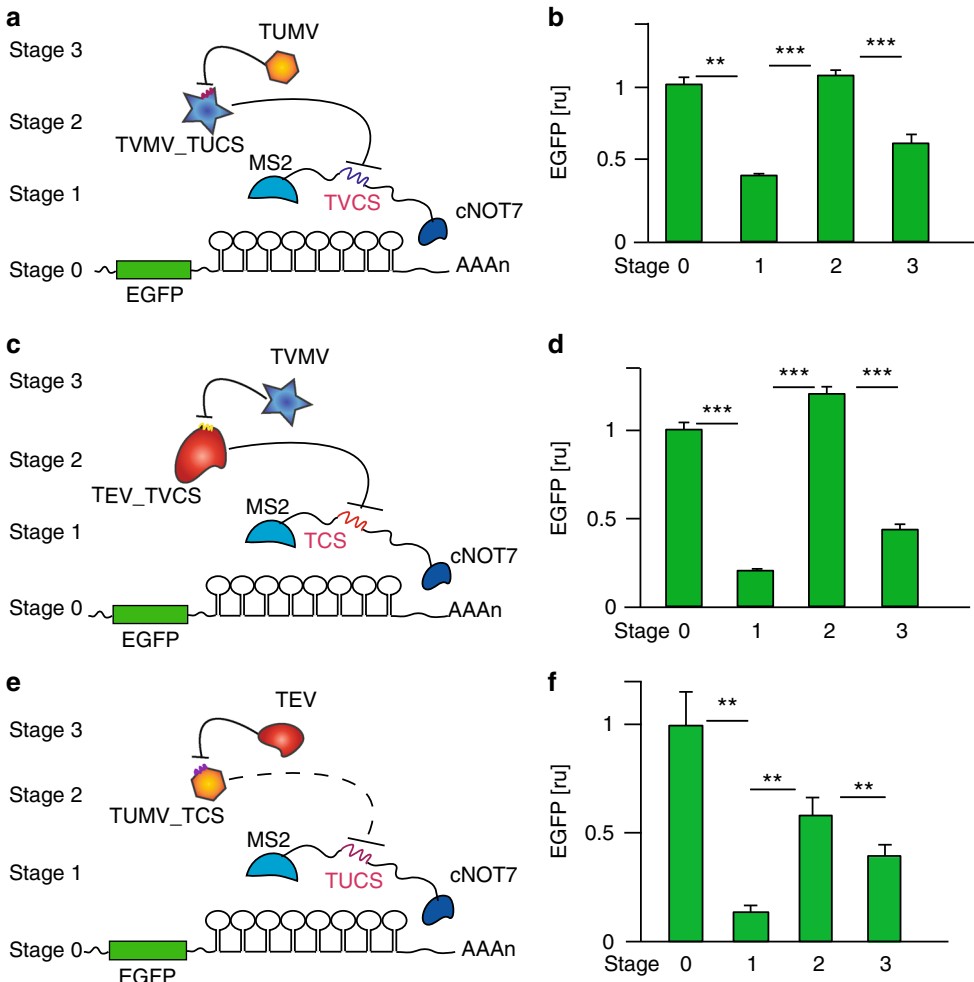

**Fig. 4** Protease–protease systems. **a**, **c**, **e** Schematics of the three-stage signaling cascades. EGFP translation is repressed by MS2-CS-cNOT7 (Stage 1), while RBP activity is disrupted by a protease (Stage 2), which is itself engineered to include a cleavage site for an orthogonal protease. When Stage 2 protease activity is inhibited by an upstream protease (Stage 3), MS2-CS-cNOT7 is able to repress target mRNAs. **b**, **d**, **f** Corresponding flow cytometry data for the geometric mean and standard deviation of EGFP normalized by transfection marker mKate and rescaled to no repression values. ru relative units. Significant changes determined by unpaired $t$-test are indicated with asterisks ***$p$-value < 0.001, **$p$-value < 0.005 $n = 3$ replicates. Data collected 48 h post transfection

to the native protein that responds to small molecules. These domains, which are stabilized by externally provided small molecules, enable fine tuning of protein stability and thus control of abundance[50–54]. Alternatively, destabilizing elements can be added in the 3'UTR of the mRNA encoding the protein of interest such as AU-rich motifs[54–56]. Engineered L7Ae based devices was programmable both in HEL293FT and HeLa cell lines, although the level of repression was reduced in HeLa cells. This confirm that HEK293FT cells are the optimal cell line for design and test synthetic biology-based devices for high transfection rate, and production capacity. Different behavior is perhaps due to different cellular composition and resourced distribution that may feedback on circuits performances. Identifying the bottle neck of circuits functionality could perhaps help to optimize genetic circuits design.

Similar to L7Ae, engineering of the proteases was based on crystal structure analysis of TEVp and TVMVp, and we observed that the best insertion site was in a similar region of these two proteases. We confirmed the same result for TUMVp, whose crystal structure has not been resolved yet, but was inferred with a homology-based model[41]. For adding new regulatory elements, computational protein design methods are becoming increasingly

accurate at predicting folding and functionality of newly design proteins[57–60], which should facilitate the construction of new protein-based synthetic networks. We envision that our platform could also be connected to intracellular protease inputs (i.e., caspases) without requiring modifications to endogenous pathways to link changes of intracellular state to output regulation. It has been recently demonstrated that synthetic mRNAs harboring cell-type-specific microRNAs can be used to efficiently purify human pluripotent stem cell (hPSC)-derived populations. Similarly, we envision that protease-responsive RBPs may represent an effective means to link detection of specific cellular states such as infection or cancer to conditional expression of output genes. For example, the SUMO1-/sentrin-specific protease SENP1 has a pro-oncogenic role in several types of cancer (including pancreatic ductal adenocarcinoma (PDAC), prostate cancer, hepatocellular carcinoma, and metastatic neuroblastoma tissues), correlating with poor prognosis[61–63]. Circuits such as ours that sense SENP1 can be used to reprogram cancer cells to express relevant immunomodulators, or trigger specific cell killing by modifying circuit output in an application-tailored fashion. Overall, our system provides a strategy for composable and scalable circuit design encoded post-transcriptionally in

mammalian cells, offering a useful research toolbox with potential benefits for RNA delivered in vivo applications such as cell-based therapies, vaccination, regenerative medicine, and biotechnological applications.

## Methods

**Cell culture.** HEK293FT (Invitrogen) and HeLa cells used in this study were maintained in Dulbecco's modified Eagle medium (DMEM, Cellgro) supplemented with 10% FBS (Atlanta BIO), 1% penicillin/streptomycin/L-Glutamine (Sigma-Aldrich) and 1% non-essential amino acids (HyClone) at 37 °C and 5% $CO_2$.

**Transfection and fluorescence imaging.** Transfections were carried out in 24-well plate format. L7Ae-CS-based devices transfections were carried out with Attractene (Qiagen). An aliquot of 400 ng total DNA was mixed with DMEM base medium (Cellgro) without supplements to a final volume of 60 μl. A total of 1.5 μl attractene was added to the dilutions and the samples were promptly vortexed to mix. The complexes were incubated for 20–25 min During the incubation time, cells were harvested by trypsinization and $2 \times 10^5$ cells seeded in 500 μl of complete culture medium in 24-well plate. Transfection complexes were added dropwise to the freshly seeded cells, followed by gentle mixing. Cells were supplemented with 1 ml of fresh growth medium 24 h post transfection and analyzed by flow cytometry after 48 h.

MS2-cNOT7 devices, L7AeCS3 experiment in HeLa cells and 96 h experiments were performed with Lipofectamine 3000 following manufacturer instructions. Total of up to 400 ng DNA was mixed with Opti-MEM I reduced serum medium (Life Technologies) to a final volume of 100 μl followed by addition of 0.5 μl P3000 reagent. After 5 min, 1.5 μl Lipofectamine 3000 was added, the samples were briefly vortexed and incubated for 30 min at room temperature. During the incubation time, $2 \times 10^5$ cells were harvested by trypsinization and seeded in 500 μl of complete culture medium in 24-well plate. Transfection complexes were added dropwise to the freshly seeded cells followed by gentle mixing. Transfection tables that list components and relative concentrations used in this study are available in the Supplementary Information (Supplementary Table 3). Fluorescence imaging and bright-field micrograph were acquired with Evos Cell Imaging System (Life Technology), using ×10 objective.

**Flow cytometry and data analysis.** Cells were analyzed with LSR Fortessa flow cytometer, equipped with 405, 488, and 561 nm lasers (BD Biosciences). We collected 30,000–100,000 events per sample and fluorescence data were acquired with the following cytometer settings: 488 nm laser and 530/30 nm bandpass filter for EYFP/EGFP, 561 nm laser and 610/20 nm filter for mKate, and 405 nm laser, 450/50 filter for EBFP. Data of 96 h experiment for L7AeCS3 and MS2-CS-cNOT7 devices and L7AeCS3 test in HeLa cells were collected with BD Accuri C6 (BD Biosciences). Population of live cells was selected according to FCS/SSC parameters. Data analysis was performed with Flowjo. Red fluorescent protein (mKate) was used in all experiments as transfection marker. For each sample, we gated the population of live cells and then the mKate-positive cells including the frequency. Whitin the mKate-positive population we calculated the geometric mean (Geo-Mean) of mKate, and the Geo-mean and frequency of EGFP, EYFP, and EBFP positive cells[64]. For L7AeCS-based devices the plotted data represent the ratio between reporter Geo-mean and frequency and mkate Geo-mean and frequency: EGFP values (EGFP Geo-Mean*%EGFP + ) normalized by the transfection marker (mKate Geo-Mean*%mKate + ) for all mKate-positive cells. For MS2-CS-cNOT7-based devices, data represent the ratio between EGFP Geo-mean and mKate Geo-mean.

Plots in Figs. 1c, 3b, d, and 4b, d, f, represent fold change values of reporter expression using an experimental condition as reference (1). The error of fold change is calculated as follow:

$$\sqrt{\left(\frac{mB}{mA}\right)^2 \times \left[\left(\frac{sA}{mA}\right)^2 + \left(\frac{sB}{mB}\right)^2\right]}$$

where mA is mean A, mB is mean B, sA is the standard error associated with mA and sB is the standard error associated with mB. Data collected for analysis 48 h after transfection.

**DNA cloning and plasmid construction.** Plasmid vectors carrying gene cassettes were created using In-Fusion HD cloning kit (Clonetech), Gateway system (Life technology), or Golden Gate system.

Reaction included 1:2 molar ratio of plasmid backbone:gene insert starting with 100 ng of vector backbone digested with selected restriction enzymes.

All plasmids used in this study consist of a constitutive promoter driving the gene of interest. The only exception is p37-pEXPR-2_3-TRE_tight-TEVp-LD0-scFv162, which expression induced by doxycycline in the presence of rtTA3. pL-A1 (pT-GTW6-CMV-mKate)[12] was used as destination vector for L7AeCS, MS2-CS-cNOT7 and protease engineering. L7Ae cleavage sites were synthesized as gblocks

(IDT). MS2 and cNOT7 and proteases were amplified with primers including proteases cleavage sites (primers are listed in the Supplementary Table 2) using the Accuprime PFX DNA polymerase (ThermoFischer Scientific), and In-Fusion reactions were performed with 1:2:2 molar ratio (vector:PCR1:PCR2). Plasmids were confirmed by sequencing analysis. 2Kt-EBFP and 2Kt-TEVp-2A-EYFP-4xFF4 plasmids were generated starting from pL-S1[12] using In-Fusion cloning.

## Data availability

The source data for all figures of this study are available from the corresponding author upon reasonable request. Plasmid sequences have been deposited at GenBank under the accession codes found in Supplementary Table 1.

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

## Acknowledgements
We thank Jesus Fernandez-Rodriguez for providing TVMV, SuMMV, and TUMV protease sequences. We thank Diego di Bernardo for critical reading of the manuscript. We thank Barbara Tumaini (Telethon Institute of Genetics and Medicine, Naples) for helping with flow cytometry data collection for experiment in HeLa cells and 96 h experiment in HEK293FT cells. We thank Annamaria Carissimo and Claudia Angelini (IAC-CNR, Naples) and Filippo Menolascina (University of Edinburgh) for providing guidance on statistical analysis. This work was supported by the Istituto Italiano di Tecnologia, the Imperial Research Fellowship and NIH P50 GM098792.

## Author contributions
V.S. conceived the idea, designed and performed experiments, and carried out data analysis. F.C. designed and performed experiments, and data analysis of engineered proteases. V.S. and F.C. built the devices. L.W. performed computational analysis on protein structure to indicate insertion points. V.S. wrote the manuscript. R.W., L.W., and F.C. edited the manuscript.

## Additional information

**Competing interests:** The authors declare no competing interests.

