## [Peer Review File · Nature Communications]

Reviewers' Comments:

Reviewer #1:

Remarks to the Author:

In this paper, Cella et al aim at expanding the toolbox of post-transcriptional regulators available for Synthetic Biology when RNA-only devices are used for circuit design in mammalian cells. Authors start with employing RBPs that have already been used in previous work (Wroblewska et al. Nat Biotech 2015) and engineer them to be responsive to protease regulation. Based on this idea, authors prove that protease regulation can be exploited to regulate the activity of RNA-binding proteins and of other proteases in order to build complex RNA-based circuits and functionalities.

I consider the idea interesting and of possible utility for the Synthetic Biology community looking at novel and faster ways to engineer mammalian cells. I therefore recommend the publication of the paper.

I have some considerations that if addressed I believe could make the paper more appealing and of broader interest:

1) Authors strongly claim that RNA-based systems represent potential in vivo therapeutics. However, in the paper, functionality of the proposed circuits is shown only for a single time point at 48 hours after transfection. I think showing a longer time-frame is needed to prove robustness, stability and the possible efficacy of the systems as therapeutic tools. In this respect, authors should also discuss how the systems could be employed in therapy considering the short-term stability of transient transfections and RNA molecules: which kind of applications do the authors have in mind?

2) Even if interesting, data are shown only for a single cell line. Can authors justify why the HEK293FT cell line has been chosen? This cell line is usually preferred when high-titer lentivirus production is the goal. Since authors claim the new regulators will be of use for RNA-based cell engineering, can they show functionality and robustness of the developed systems also in other cellular systems? I believe this is needed to prove the importance potential novel tools developed with broad application in mind.

Minor points:

1) No indication of the time of measurement is present in the figures; I suggest authors add this information.

2) Writings are very small in the figures at times when looking at the manuscript, especially in the construct diagrams (e.g. Figure 2e and Figure 3b). I would suggest improving writing size.

Reviewer #2:

Remarks to the Author:

Cella and colleagues present the proof-of-principle for the post-transcriptional regulation of mammalian gene expression based on RNA binding proteins engineered to be susceptible to viral proteases. The work builds on previous efforts by the team (ref. 16) and parallels previous work carried out in microbes (ref. 18).

The authors' long-term goal is the development of RNA-based regulatory circuits and this is a small step towards it: the work demonstrates that engineered protease regulation of gene expression is possible. Although there is strong emphasis on the delivery of RNA/protein complexes, the work follows the more traditional route of transfecting mammalian cells with DNA for development and validation of the circuits. Multiple RNA binding proteins and proteases are engineered to introduce viral protease target sites, validated for protease inactivation, and different regulatory circuit topologies are demonstrated at the population level. The work is an

addition to the field and may provide a platform for the optimization and characterization of RNA/protein complexes delivery.

I have some suggestions to improve the manuscript which I trust the authors will address to enable publication. They are roughly in the order they occur in the manuscript, but more important suggestions are highlighted with *.

1. * Introduction – it would be useful to clarify the experimental setup in the introduction. The opening emphasis on RNA/protein circuits can be misleading when the work is still reliant on DNA-based circuits post-transcriptionally regulated. I, for one, initially thought the work was being done cell-free.

2. Line 106-107 – de-repression or derepression?

3. Figure 1a and 1b can be combined.

4. Figure 1 – The website link to the L7Ae structure can probably be replaced simply by the PDB Id.

5. * Figure 1d and all subsequent ones – The data have to be presented as box plots rather than bar plots. As shown in all flow data provided by the authors, the populations are not simple log-normal distributions (with some as in Figure 3b clearly bimodal). In that context, bar plots are misleading.

6. * Figure 1d and subsequent ones – Although a statistical relevance value is given, there is no description of which test was used and no explanation to justify the validity of the test for the given population.

7. Figure 1f – The signals in the HCV sensor are very weak. Are they significant? What fraction of the population are raising a significant response (which may be a more relevant measure here)? How close is that circuit to a viable biosensor?

8. *M&M – Experimental detail is sparse and must be clarified. How are the circuits being induced? Is there any evidence for the levels of expression of repressors, NS3 or intrabodies?

9. Logic gate described in L.136 and Figure 2B – I think an “OR NOT” gate is confusing, and the pictorial is not standard. I think clearer would be to describe it as a NOT gate feeding as one of the inputs into an OR gate. Similarly, the main text could be altered to: “... with two input logic, based on presence of siRNA-FF5 and the absence of siRNA-FF4, that operates...”

10. * EBFP repression (L.145) – I think that SI Figure 2 or 5 would be more effective ways of presenting the data in Figure 2b. In fact, I would also suggest the authors gate some of the populations to allow better characterisation of the population – is this a population response with large cell-to-cell variation or a consistent cell response? While data in Figure SI5 suggests the latter, there is not enough information on the flow cytometry M&M for me to be able to confirm that. A consistent cellular response is crucial if this platform is to be used for developing biosensors. Also, it is not explicit from figure or text if the populations are statistically different.

11. Minor spelling errors: wild-type (L.164), SWISS-MODEL (L.195), transfected (M&M L.99)

12. MS2-TCS-CNOT7 activation (L.164 and Figure 3b) – It should be made clearer that TEV was induced in a MS2-TCS-cNOT7 background. Also, it is not clear that the MS2-cNOT7 control has been carried out in the presence of TEV - that would be the desirable control.

13. Figure 3e – colour scale is too subdued. It would be more appropriate to use a stronger

contrasting colour (e.g. white). However, while this approach gives a very clear, visually appealing, summary, is the geometric mean of a non-(log)normal population the appropriate summarising statistic?

14. Figure 3d – MS2-CS-CNOT7 repression seems significantly weaker for proteases between Figure 3b and 3d (and SI Fig 6~9). Were the normalizations carried out differently between the assays? If no difference was expected, could the authors elaborate on the potential reasons for such difference?

15. Results on the engineered protease circuit L.178~L.195 - I think this section could be clearer if a different structure was followed, maybe presenting the concept, then the protease engineering, then bringing it back to the experiment at hand. As is, it took me a couple of reads to understand the experiment.

16. Figure 4 (b,d,f) – It is not clear from text or figure legend what the experiments described here are. I presume (e.g. 4b) that they are additive, starting without repression, introducing repression (MS2-TVCS-cNOT7), alleviating repression (MS2-TVCS-cNOT7 + TVMV_TUCS2) and reintroducing it via protease control (MS2-TVCS-cNOT7 + TVMV_TUCS2 + TUMV). Again (point #8), there is no M&M information on how this experiment was carried out, how constructs were induced, no experimental times. As is, it would not be possible to reproduce the experiments from the information given.

Reviewer #1 (Remarks to the Author):

In this paper, Cella et al aim at expanding the toolbox of post-transcriptional regulators available for Synthetic Biology when RNA-only devices are used for circuit design in mammalian cells. Authors start with employing RBPs that have already been used in previous work (Wroblewska et al. Nat Biotech 2015) and engineer them to be responsive to protease regulation. Based on this idea, authors prove that protease regulation can be exploited to regulate the activity of RNA-binding proteins and of other proteases in order to build complex RNA-based circuits and functionalities. I consider the idea interesting and of possible utility for the Synthetic Biology community looking at novel and faster ways to engineer mammalian cells.

I therefore recommend the publication of the paper. I have some considerations that if addressed I believe could make the paper more appealing and of broader interest:

We thank the reviewer for the positive comments. Below we address the reviewer's considerations.

1) Authors strongly claim that RNA-based systems represent potential *in vivo* therapeutics. However, in the paper, functionality of the proposed circuits is shown only for a single time point at 48 hours after transfection. I think showing a longer time-frame is needed to prove robustness, stability and the possible efficacy of the systems as therapeutic tools. In this respect, authors should also discuss how the systems could be employed in therapy considering the short-term stability of transient transfections and RNA molecules: which kind of applications do the authors have in mind?

Response: We thank the reviewer for the insightful comment. RNA is certainly a powerful regulator of cellular functions, and as a result, ongoing research is focusing intently on RNA-based therapeutic strategies. A key issue in this regard is RNA stability; indeed chemical and structural modifications of *in vitro* transcribed mRNA that enhance its stability have enabled the *in vivo* production of significant levels of target proteins for durations longer than a week (Holtkamp et al 2006; Kariko et al 1999; Kallen et al 2014; Sahin et al 2014). Recently, self-replicating RNA viruses such as Sindbis have gained interest for a wide range of medical applications including vaccine delivery, gene therapy and cellular reprogramming (Lundstrom et al 2012; Yoshioka et al 2013). A main advantage of RNA replicons is that they self-replicate and hence can generate high levels of gene products even when starting from a relatively low dose, while the risk of undesired integration is minimal because they do not reverse transcribe the RNA into DNA (Strauss et al 1994; Dryga et al 1997). Moreover, Sindbis replicons with reduced cytopathicity and even longer-term expression have been developed, making them an attractive platform for engineering RNA-encoded synthetic devices (Dryga et al 1997; Frolov et al 1999; Heise et al 2003; Perri et al 2000; Beal et al 2014).

RNA-based regulatory devices can also be used for applications that do not require long and sustained mRNA translation. Miki et al (Miki et al 2015) demonstrated that synthetic RNAs can be used to efficiently purify human pluripotent stem cell (hPSC)-derived populations. In their system synthetic mRNAs encode a fluorescent protein tagged with microRNA (miRNA) target sites specific for the cell of interest. These miRNA switches efficiently purified cardiomyocytes, hepatocytes, endothelial cells and insulin-producing cells upon differentiation from hPSCs, overcoming isolation strategies issues. We envision that devices based on protease-RBPs synthetic regulation could be used to link expression of the output gene to a specific cellular state such as viral infection or cancer. For example, the SUMO1/sentrin specific protease SENP1 has been shown to have a pro-oncogenic role in many types of cancer, correlating with poor prognosis. SENP1 is upregulated in pancreatic ductal adenocarcinoma (PDAC), prostate cancer, hepatocellular carcinoma and metastatic neuroblastoma tissues (Ma et al 2014; Cui et al

2016; Xiang et al 2016). This may enable construction of RNA-encoded vaccines that induce antigen expression only in specific cancer cells, or protease-RBPs based sensor-actuator, where input SENP1 disrupts RBP function and the output is then a proapoptotic gene. We have included this discussion, with appropriate references, in the revised version of the manuscript (L208-220 and 256-267).

In addition to the expanded discussion, we also investigated the robustness of the system for longer time frame as suggested by the reviewer. We tested the L7AeCS3 and MS2-TCS-cNOT7 devices (Figure 1b-c, Figure 3b) and collected data 96h post transfection. We observed that our system is stable over longer time-frame. L7AeCS3 data are reported in **Supplementary Figure 4**. MS2-CS-cNOT7 data are reported in **Supplementary Figure 10**. We modified the main text to reflect these observations (L98-100; L164-166).

2) Even if interesting, data are shown only for a single cell line. Can authors justify why the HEK293FT cell line has been chosen? This cell line is usually preferred when high-titer lentivirus production is the goal. Since authors claim the new regulators will be of use for RNA-based cell engineering, can they show functionality and robustness of the developed systems also in other cellular systems? I believe this is needed to prove the importance potential novel tools developed with broad application in mind.

Response: We thank the reviewer for this comment. We chose HEK293FT for our studies since, in addition to HEK293T and HEK293 cells, these cells are a commonly used platform to test synthetic circuits. An important advantage of these cells is their high transfection efficiencies, guaranteeing high protein production levels when using transfection reagents at reduced costs in comparison to other hard-to-transfect cells. This is especially important when testing libraries of synthetic circuits with transient transfections. Nonetheless, we appreciate the reviewer's suggestion, and therefore also opted to test the L7AeCS3 repression-derepression system in HeLa cells. L7AeCS3 showed repression of EGFP albeit with lower efficiency as compared to HEK293FT, whereas TEVp was able to restore EGFP expression. We report the new data in new **Supplementary Figure 5**. We modified the main text accordingly (L98-100).

Minor points:

1) No indication of the time of measurement is present in the figures; I suggest authors add this information.

2) Writings are very small in the figures at times when looking at the manuscript, especially in the construct diagrams (e.g. Figure 2e and Figure 3b). I would suggest improving writing size.

Response: We thank the reviewer for these suggestions. We have now added the time of data collection analysis both in the caption of the main figures and in the Materials & Methods Section. We have also modified the font sizes in the figures to make them more legible.

Reviewer #2 (Remarks to the Author):

Cella and colleagues present the proof-of-principle for the post-transcriptional regulation of mammalian gene expression based on RNA binding proteins engineered to be susceptible to viral proteases. The work builds on previous efforts by the team (ref. 16) and parallels previous work carried out in microbes (ref. 18).

The authors' long-term goal is the development of RNA-based regulatory circuits and this is a small step

towards it: the work demonstrates that engineered protease regulation of gene expression is possible. Although there is strong emphasis on the delivery of RNA/protein complexes, the work follows the more traditional route of transfecting mammalian cells with DNA for development and validation of the circuits. Multiple RNA binding proteins and proteases are engineered to introduce viral protease target sites, validated for protease inactivation, and different regulatory circuit topologies are demonstrated at the population level. The work is an addition to the field and may provide a platform for the optimization and characterization of RNA/protein complexes delivery.

I have some suggestions to improve the manuscript which I trust the authors will address to enable publication. They are roughly in the order they occur in the manuscript, but more important suggestions are highlighted with *.

Response: We thank the reviewer for the positive feedback.

1. * Introduction – it would be useful to clarify the experimental setup in the introduction. The opening emphasis on RNA/protein circuits can be misleading when the work is still reliant on DNA-based circuits post-transcriptionally regulated. I, for one, initially thought the work was being done cell-free.

Response: We thank the reviewer for this helpful comment. To clarify the experimental setup, we have now added a sentence in the introduction (L68-69): *"The devices were genetically encoded on plasmid DNA to allow rapid testing in human embryonic kidney 293 (HEK293) cells via transient transfection."*

2. Line 106-107 – de-repression or derepression?

Response: Thank you. We fixed to derepression.

3. Figure 1a and 1b can be combined.

Response: We combined these two figure panels.

4. Figure 1 – The website link to the L7Ae structure can probably be replaced simply by the PDB Id.

Response: Based on this recommendation, the PDB id is now included in the revised version of the manuscript.

5. Figure 1d and all subsequent ones – The data have to be presented as box plots rather than bar plots. As shown in all flow data provided by the authors, the populations are not simple log-normal distributions (with some as in Figure 3b clearly bimodal). In that context, bar plots are misleading.

Response: We thank the reviewer for this comment. While we agree that box plots provide information related to EGFP distribution within a single experiment, bar charts are the standard method used to represent the average of normalized geometric means and standard deviation computed for several experimental replicates (Wroblewska et al *Nature Biotechnology* 2015; Kiani et al *Nature Methods* 2015; Stanton et al *ACS Synthetic Biology* 2015; Siciliano et al *Nature Communications* 2018; Yeo et al *Nature Methods* 2018). We therefore believe that bar charts provide the appropriate summary statistic to represent our replicate data suitable for the main narrative. To provide the readers with a thorough depiction of the actual distributions in our experiments, we include flow cytometry histograms and dot plots in the Supplementary Information (current SF2 and SF8), and moved the original representative histograms in Figure 3b to Supplementary Information (**Supplementary Figure 9**) for easier comparison.

6. Figure 1d and subsequent ones – Although a statistical relevance value is given, there is no description of which test was used and no explanation to justify the validity of the test for the given population.

Response: Thank you for raising this important point. We used a paired T-Test comparing the normalized geometric means for the different experimental conditions. For example, in Figure 1c, paired t-test was calculated from the normalized GeoMean of dEGFP in the presence and absence of TEVp. We have now added to the caption of each of the relevant figures the following sentence: “*Significant changes determined by paired t-test are indicated with asterisks.*”

7. Figure 1f – The signals in the HCV sensor are very weak. Are they significant? What fraction of the population are raising a significant response (which may be a more relevant measure here)? How close is that circuit to a viable biosensor?

Response: We thank the reviewer for the comment. We agree that for this particular experiment, we only observe a fold change of 2x. In principle, optimizing the circuit further could be attempted with different NS3 binding intrabodies, or perhaps different flexible linkers in the chimeric proteins. In this specific configuration, one possibility for the reduced performance relative to the other sensors is that binding to NS3 does not allow full access of TEVp to the cognate cleavage site. However, given constraints on experimental efforts and the focus on other elements in the manuscript, we did not investigate alternative variants and sensor configurations. However, to address the reviewer comment, we performed a paired t-test on the two populations with the two alternative L7AeCS3 fused to the N or C-terminus of scFv35 intrabody, and added the significance evaluation where appropriate, as well as an updated Figure 1e caption.

8. M&M – Experimental detail is sparse and must be clarified. How are the circuits being induced? Is there any evidence for the levels of expression of repressors, NS3 or intrabodies?

Response: We understand the reviewer concern and now provide detailed transfection tables for the experiments performed in this study, reporting on plasmids used, concentration of each component and the transfection marker (**Supplementary Table 1**). All circuits components are induced by constitutive promoters and transiently expressed in HEK293FT cells, whereas in the NS3 sensor (Fig. 1e) TEVp-scFv162 expression is transcriptionally regulated by Doxycycline-inducible TRE promoter (we have now added this information in the main text and in the M&M section). To test the circuits, we prepared several liposome-plasmids complexes that included all the experimental conditions. For example, in Figure 4 each stage includes the addition of the new components (Stage 0: plasmid encoding EGFP. Stage 1: plasmid encoding EGFP *plus* plasmid encoding MS2-CS-cNOT7). For all the transfections performed, we use transfection marker mKate as an indication of successful transfection and co-expression of the components. This is a commonly used procedure for multiple plasmid transfections in mammalian cells. Moreover, NS3 protein was tagged with a blue fluorescent tag, and therefore its expression could be followed directly in our experiments. In this respect, we added a sentence in the Results section (L124-127): “*Next, we tested the HCV sensor in HEK293 cells by expressing scFv35-L7Ae-CS3 or L7Ae-CS3-scFv35 along with TEVp-scFv162, in the presence or absence of a blue fluorescence-tagged NS3 (BFP-nNS3)²⁰, useful for live cell monitoring of target protein expression.*”

9. Logic gate described in L.136 and Figure 2B – I think an “OR NOT” gate is confusing, and the pictorial is not standard. I think clearer would be to describe it as a NOT gate feeding as one of the inputs into an OR gate. Similarly, the main text could be altered to: “... with two input logic, based on presence of siRNA-FF5 and the absence of siRNA-FF4, that operates...”

Response: We appreciate the reviewer suggestion and we provide in the revised manuscript full detailed explanation of the logic circuit. The actual logic function is computed using the following logic simplification process:

$$\begin{aligned} \text{EBFP} &= \text{NOT} (\text{L7Ae}) \\ \text{L7Ae} &= (\text{NOT} (\text{siRNA-FF5})) \text{ AND } (\text{NOT TEVp}) \\ \text{TEVp} &= (\text{NOT siRNA-FF4}) \end{aligned}$$

Coalescing these terms yields:

$$\text{EBFP} = \text{NOT} ((\text{NOT} (\text{siRNA-FF5})) \text{ AND } (\text{NOT} (\text{NOT siRNA-FF4})))$$

Then, with elimination of NOT/NOT, it can be simplified to:

$$\text{EBFP} = \text{NOT} (\text{NOT} (\text{siRNA-FF5})) \text{ AND } \text{siRNA-FF4}$$

Further simplification using DeMorgan's law, where $\text{NOT} (a \text{ AND } b) = (\text{NOT } a) \text{ OR } (\text{NOT } b)$:

$$\text{EBFP} = (\text{NOT} (\text{NOT} (\text{siRNA-FF5}))) \text{ OR } (\text{NOT siRNA-FF4})$$

And again, elimination of NOT/NOT, to yield the final simplified logic function:

$$\text{EBFP} = \text{siRNA-FF5 OR } (\text{NOT siRNA-FF4})$$

We modified the main text of the manuscript: “a two-input three-stage repression cascade with an overall logic function that is true in the presence of siRNA-FF5 or absence of siRNA-FF4”, and added a new supplementary figure that explains the simplified logic circuit scheme (**Supplementary Figure 7**)

10. EBFP repression (L.145) – I think that SI Figure 2 or 5 would be more effective ways of presenting the data in Figure 2b. In fact, I would also suggest the authors gate some of the populations to allow better characterisation of the population – is this a population response with large cell-to-cell variation or a consistent cell response? While data in Figure SI5 suggests the latter, there is not enough information on the flow cytometry M&M for me to be able to confirm that. A consistent cellular response is crucial if this platform is to be used for developing biosensors. Also, it is not explicit from figure or text if the populations are statistically different.

Response: We are certainly interested in providing as much information as possible for our experiments. In this particular case, the small magnitude of the standard deviations, as shown in Figure 2b, clearly supports the observation that the experimental results are very consistent for each particular condition tested. As mentioned in comment #5 above, the bar charts included here depict the average of geometric means for each output, as normalized by transfection marker mKate for three experimental replicates. We believe that this analysis provides very useful information about the consistent behavior of the cellular population across the experiments performed, and is in line with previous publications in the field (e.g. Wroblewska et al *Nature Biotechnology* 2015). In SF8 we also provide the dot plots themselves for those who wish to inspect the actual distributions. Gating on the subpopulations, e.g. measuring the percentage of cells above a particular threshold, could in principle provide further information about how the population distributions change. But in this case, we do not believe it is actually necessary to report this additional information, as the results are pretty clear already. With respect to the significance of the differences in the outputs for the different conditions, we have performed a paired t-test analysis and added information about statistical significance where appropriate. We modified the caption of Figure 2 accordingly.

11. Minor spelling errors: wild-type (L.164), SWISS-MODEL (L.195), transfected (M&M L.99)

Response: we thank the reviewer for catching these typos, which are now fixed.

12. MS2-TCS-CNOT7 activation (L.164 and Figure 3b) – It should be made clearer that TEV was induced in a MS2-TCS-cNOT7 background. Also, it is not clear that the MS2-cNOT7 control has been carried out in the presence of TEV - that would be the desirable control.

Response: We have now modified the sentence in L.164 (now L163-166) to specify that TEV protease was co-expressed with MS2-TCS-CNOT7, thus inducing EGFP derepression, as follows: “*In absence of TEVp, MS2-TCS-CNOT7 showed similar repression to its wildtype counterpart, whereas co-expression of the protease fully restored EGFP expression, indicating efficient and sustained disruption of RBP activity.*” As the reviewer suggested, we also performed a further control experiment to evaluate the behavior of wild type MS2-cNOT7 in presence of TEVp. As anticipated, our results indeed indicate that TEVp does not exhibit any activity on wild type MS2-cNOT7. This result is now part of **Supplementary Figure 10** where we tested the behavior of the circuits for 96h, as suggested by Reviewer #1, comment #1. We modified the main text accordingly (L166-167).

13. Figure 3e – colour scale is too subdued. It would be more appropriate to use a stronger contrasting colour (e.g. white). However, while this approach gives a very clear, visually appealing, summary, is the geometric mean of a non-(log)normal population the appropriate summarising statistic?

Response: We have modified the heatmap as suggested by the reviewer. For the discussion about summary statistics, please refer to our response to comment #5.

14. Figure 3d – MS2-CS-CNOT7 repression seems significantly weaker for proteases between Figure 3b and 3d (and SI Fig 6~9). Were the normalizations carried out differently between the assays? If no difference was expected, could the authors elaborate on the potential reasons for such difference?

Response: We understand this concern, which is likely due to potentially unclear descriptions of Figures 3b and 3d. In Figure 3b, we plot the relative EGFP levels when normalized to the baseline condition of non-repressed negative control (NC), where only the EGFP reporter is present, with this level designated as 1. In Figure 3d, we report the repression/derepression of EGFP by MS2-CS-CNOT7 variants that include cleavage sites responsive to SuMMVp, TVMVp and TUMVp. For each protease and cognate cleavage site, the minus (-) condition indicates full repression of EGFP by MS2-CS-CNOT7 in absence of the protease, whereas the plus (+) condition indicates the derepression in presence of the protease. We normalize full repression strength to be designated with a value of 1. As such, by normalizing to the full repression condition, we plot the fold change between the fully repressed state and derepression via inactivation of the particular MS2 variant.

To allow an easier comparison between the proteases used in this system, we modified figure 3d and designated full repression by MS2-CS-cNOT7 as 1. We then reordered in Figure 3d the proteases from most efficient to least in terms of derepression of EGFP, meaning TUMV-> TVMV->SuMMV. In terms of EGFP repression (NC/Rep), from most efficient to least: SuMMV > TVMV > TUMV > TEVp. We modified the main text to reflect these changes (L173-176).

15. Results on the engineered protease circuit L.178~L.195 - I think this section could be clearer if a different structure was followed, maybe presenting the concept, then the protease engineering, then bringing it back to the experiment at hand. As is, it took me a couple of reads to understand the experiment.

Response: We appreciate the reviewer suggestion. To make this section more clear, we followed the advice provided here by the reviewer, and moved a portion of Supplementary Note 1 to the main text,

thereby first describing the rational design of the engineered proteases. We believe that this will help readers better understand the logical and experimental flow of the section.

16. Figure 4 (b,d,f) – It is not clear from text or figure legend what the experiments described here are. I presume (e.g. 4b) that they are additive, starting without repression, introducing repression (MS2-TVCS-cNOT7), alleviating repression (MS2-TVCS-cNOT7 + TVMV_TUCS2) and reintroducing it via protease control (MS2-TVCS-cNOT7 + TVMV_TUCS2 + TUMV). Again (point #8), there is no M&M information on how this experiment was carried out, how constructs were induced, no experimental times. As is, it would not be possible to reproduce the experiments from the information given.

Response: To address this reviewer concern, we have added to Figure 4 the various stages of the protein-protein cascade, and included as supplementary tables the transfections performed. These tables describe in detail the conditions used in these experiments.

Reviewers' Comments:

Reviewer #1:

Remarks to the Author:

I believe the revised version of the manuscript by Cella et al is now an improved version of the paper.

I appreciate authors` efforts to show the system`s functionality at longer time points and in different genetic backgrounds.

I believe the paper is now suitable for publication.

Just one minor comment: data reported in Supplementary Figures 5 and 10 show high background levels for the systems in the OFF-state. Could the authors comment on this in the discussion of the manuscript? Could this be a potential drawback in the application of the developed constructs as tools for broader use in synthetic biology? Which future work could help improving this?

Reviewer #2:

Remarks to the Author:

I thank the authors for addressing most of the points I raised in my initial assessment of the manuscript. A couple of points remain outstanding regarding the statistical analysis and presentation of data (numbering as per original report):

5. Presentation of non-normal distributions as bar graphs. As per my original point, bar graphs of non-normal distributions are misleading statistics. If that has been accepted in previous publications, that reflects the expertise and standards of previous referees. I leave at the discretion of the editor as to whether this needs to be addressed prior to publication.

6. Statistical test. Thank you for clarifying which test has been used to compare samples (paired t-test). t-tests assume an underlying normal distribution (or normality of a log-transformed log-normal distribution). As long as the null hypothesis is based on a normal distribution, which based on the provided typical flow cytometry results should hold, a t-test is appropriate. I leave to the authors to confirm that null hypotheses were always against log-normal populations.

Nevertheless, given experimental setup, a paired t-test is not appropriate. The comparison is not of the same sample after it undergoes 'treatment' but of independent samples with one being 'treated'. Hence, where t-tests are appropriate, they should be unpaired. They may be one-tailed where the direction of the response can be argued a priori (which I believe will cover most cases presented here).

If the appropriate tests do not affect the authors' conclusions, I have no reservations to recommend publication of the manuscript.

A few additional minor points raised in the revision:

1. Figure 3 - it would be useful to clarify which axis refers to target and which refers to protease.
2. Minor typos: (L119) concentrations, (L138) Geo-mean, (L140) condition
3. L147 - (developed in-house) is unnecessary unless the authors argue that they have developed a novel methodological variant.
4. Bibliography - There are multiple mistakes, primarily incomplete details, in the current version of the bibliography.

Reviewer #1

I believe the revised version of the manuscript by Cella et al is now an improved version of the paper.

I appreciate authors` efforts to show the system`s functionality at longer time points and in different genetic backgrounds. I believe the paper is now suitable for publication.

Just one minor comment: data reported in Supplementary Figures 5 and 10 show high background levels for the systems in the OFF-state. Could the authors comment on this in the discussion of the manuscript? Could this be a potential drawback in the application of the developed constructs as tools for broader use in synthetic biology? Which future work could help improving this?

Response. We thank the reviewer for the positive feedback. We agree that background in OFF condition can be reduced with further optimization of the circuit. This could be achieved by improving the ratio of single components, or optimizing the strength of repression (e.g in the case on L7Ae one could test an increased number of binding sites in the 5'UTR of the target mRNA). As interesting perspective emerging from the different behaviour of L7Ae in HEK and HeLa cells, is that synthetic circuits performance may be cell context dependent (see for example Sedlmayer et al *Nature Communications* 2018), and optimization should be tailored to the desired cells/tissues. We believe that identifying the bottle neck of circuits functionality would help to optimize genetic circuits design. We added this comment to the discussion as suggested by the reviewer.

Reviewer #2 (Remarks to the Author):

I thank the authors for addressing most of the points I raised in my initial assessment of the manuscript. A couple of points remain outstanding regarding the statistical analysis and presentation of data (numbering as per original report):

5. Presentation of non-normal distributions as bar graphs. As per my original point, bar graphs of non-normal distributions are misleading statistics. If that has been accepted in previous publications, that reflects the expertise and standards of previous referees. I leave at the discretion of the editor as to whether this needs to be addressed prior to publication.

Response. We thank the reviewer for the feedback. However, we still think that our analysis reflects the standard of high-quality publications, and as we mentioned we do not believe it is actually necessary to report this additional information as the results are pretty clear already.

6. Statistical test. Thank you for clarifying which test has been used to compare samples (paired t-test). t-tests assume an underlying normal distribution (or normality of a log-transformed log-normal distribution). As long as the null hypothesis is based on a normal distribution, which based on the provided typical flow cytometry results should hold, a t-test is appropriate. I leave to the authors to confirm that null hypotheses were always against log-normal populations.

Nevertheless, given experimental setup, a paired t-test is not appropriate. The comparison is not of the same sample after it undergoes 'treatment' but of independent samples with one being 'treated'. Hence, where t-tests are appropriate, they should be unpaired. They may be one-tailed where the direction of the response can be argued a priori (which I believe will cover most cases presented here).

If the appropriate tests do not affect the authors' conclusions, I have no reservations to recommend publication of the manuscript.

Response. We thank the reviewer for this important comment. To verify that null hypotheses were against log-normal populations we have fitted our data with a mixture of two normal distributions, which reflects the two population of cells (non-transfected and transfected) of our experimental condition. From the density plot we observed good fit for the data used for analysis.

We also appreciate his feedback on calculating the unpaired t-test rather than the paired. We do agree that for this experimental setup unpaired t-test is more appropriate. We have carried out the 2-tail unpaired t-test, and we modified main text and figures upon proper calculations. Our conclusions were robust following unpaired t-test.

1. Figure 3 - it would be useful to clarify which axis refers to target and which refers to protease.
We clarify that in the revised figure

2. Minor typos: (L119) concentrations, (L138) Geo-mean, (L140) condition
Typos are now fixed

3. L147 - (developed in-house) is unnecessary unless the authors argue that they have developed a novel methodological variant.
We removed "developed in-house"

4. Bibliography - There are multiple mistakes, primarily incomplete details, in the current version of the bibliography.
Bibliography is now fixed